# Indoor Positioning System with UWB Based on a Digital Twin

**DOI:** 10.3390/s22165936

**Published:** 2022-08-09

**Authors:** Ping Lou, Qi Zhao, Xiaomei Zhang, Da Li, Jiwei Hu

**Affiliations:** School of Information Engineering, Wuhan University of Technology, Wuhan 430070, China

**Keywords:** ultra-wideband (UWB), indoor positioning system (IPS), NLOS error, digital twin

## Abstract

Ultra-wideband (UWB) technology is used for indoor positioning, but its positioning accuracy is usually degenerated by various obstacles in the indoor environment because of non-line-of-sight (NLOS). Facing the complex and changeable indoor environment, an indoor positioning system with UWB based on a digital twin is presented in this paper. The indoor positioning accuracy is improved with a perception–prediction feedback of cyber-physics space in this indoor positioning system. In addition, an anchor layout method with virtuality–reality interaction and an error mitigation method based on neural networks is put forward in this system. Finally, a case study is presented to validate this indoor positioning system with a significant improvement in positioning accuracy.

## 1. Introduction

Global Navigation Satellite Systems (GNSS) [1] have been extensively used in various industries and can obtain a precise location in the outdoor environment. However, due to signal attenuation caused by building occlusion, GNSS cannot provide an indoor positioning service. Indoor positioning systems (IPSs) [2] based on different signals (acoustic, visual, and radio frequency) have been widely studied. Among them, IPS based on radio frequency signals has been widely used because of its high precision and low cost.

RF signals can be divided into continuous wave (CW) and impulse signals. Indoor positioning systems using continuous waves such as WiFi [3,4], Bluetooth [5], ZigBee [6], and RFID [7] usually use fingerprint-based positioning algorithms. It is necessary to collect received signal strength indicator (RSSI) [8,9] or channel state information (CSI) [4,10] in space to build a fingerprint database. By matching the information to estimate the location of the tag, this positioning method usually has only meter positioning accuracy. The positioning system using a pulse UWB [11] signal includes an anchor and tag. A basic UWB positioning system includes multiple anchors and one tag. The position of the tag is estimated by measuring the distance or angle between the tag and multiple anchors. Common algorithms include TOF [12], TOA [13], TDOA [14], AOA [15], etc. Due to its high time resolution, the positioning accuracy of a UWB system can reach centimeter level.

Although a UWB positioning system has high positioning accuracy, in the indoor environment, the building structure and furniture may cause shielding between anchors and tags, resulting in increased ranging value and non-line-of-sight (NLOS) [3] error. Due to the complexity of the indoor environment, NLOS error exists widely. At the same time, an unreasonable anchor position will cause a wide range of shelter and aggravate the NLOS situation in the indoor environment.

A digital twin is a reality–virtual interaction technology, which helps physical entities make decisions through the interaction between physical space and digital space. In this article, an indoor positioning system (IPS) based on a digital twin with UWB signals is proposed and implemented. Based on the constructed digital twin, the optimal anchor layout, adaptive error map construction, and positioning error mitigation are achieved. Firstly, a digital space including spatial geometry and a UWB signal propagation model was constructed. Secondly, the optimal anchor layout used the slime mold algorithm (SMA) [16] to solve the optimal anchor position in digital space, guide the anchor installation in physical space, and then adjust the anchor position in digital space according to the actual installation position in physical space. The locations of the measuring points were planned in digital space according to an adaptive density. Then, the measuring points were installed in physical space to collect measurement data. After that, a neural network-based positioning error mitigation method was used to process positioning data in physical space, so as to improve the positioning accuracy of the system.

The main contributions of this article are summarized as follows:An indoor positioning system based on digital twin is proposed.The ranging error model and positioning error model of UWB positioning system were established, and the friendly visualization of the positioning error in digital space was realized.Based on the vector positioning error model, an optimal anchor layout algorithm and a positioning error mitigation method are proposed.

The remainder of this paper is organized as follows. Section 2 presents existing works related to reducing the UWB positioning error. Section 3 describes the architecture of the indoor positioning system and the associated algorithms. Section 4 describes the experimental validations of the proposed system. Section 5 concludes the paper.

## 2. Related Works

A UWB positioning system usually uses the TOA algorithm to locate and estimates the coordinates of the tag by measuring the distance between the tag and multiple anchors. The positioning accuracy can reach centimeter-level under ideal conditions because of its high time resolution. In an indoor environment, the positioning error of a UWB system mainly comes from the increase in the ranging value caused by NLOS interference. The current research mainly focuses on multisystem integration, algorithm improvement, NLOS identification, and mitigation.

Multisystem integration refers to the integration of a UWB positioning system with other positioning systems. The data of a UWB system are fused with a Bluetooth system or GNSS [17,18,19], or an inertial navigation system (INS) and radar are used to correct the data of a UWB system [20,21,22]. However, the amount of computation required for dual system fusion is large, and the system structure is complex.

Algorithm improvement can reduce the calculation error. By combining the TOA/TOF algorithm with the TDOA algorithm [23,24] or using the improved CHAN algorithm [25] or EKF/UKF/CKF algorithm [26,27,28], the solution accuracy can be improved. The method based on deep learning is used to predict the label coordinates according to the ranging value [4,29,30], but it needs longer calculation time and force requirements. Moreover, when the ranging error is significant, these algorithms cannot effectively restore the accurate positioning coordinates.

A large number of studies have focused on the identification and mitigation of NLOS errors [31,32,33]. Using median, mean, and other temporal filters [34] or an improved Kalman filter [26,27,28] can avoid the influence of a small amount of abnormal data, but it is difficult to play a role when the tag is in an NLOS condition for a long time. An NLOS error can be effectively identified by machine learning (ML) methods [29,35,36], but it is difficult to judge when the label moves. By introducing redundant anchors [37], we can filter the anchors without obstacles to participate in the calculation to improve the positioning accuracy, but it will significantly increase the system cost. It is also a feasible scheme to correct the data by constructing an error map, but it requires a large amount of measuring point data and is difficult to construct.

A digital twin is a universal technology that has emerged in recent years and is widely used in the design, manufacturing, operation, and maintenance stages of various products. Tao Fei et al. proposed a digital twin-driven fault prediction and health management method for complex equipment [38,39] and established a digital twin five-dimensional model of complex equipment [40]. At the same time, digital twin technology has also been applied in automobiles, medical care, construction, ships, etc., [41,42,43,44,45]. The application of digital twins in the field of indoor positioning can effectively reduce positioning errors and improve the efficiency of positioning system construction.

In this paper, we propose a UWB indoor positioning system using digital twin technology, which can easily migrate to other indoor scenes. A positioning error model can be constructed with only a small amount of measurement campaigns. The optimal anchor layout can be obtained quickly through the optimal anchor selection algorithm, which significantly reduces the NLOS situation. A deep learning-based error mitigation method can effectively reduce the positioning error of a UWB system and improve the positioning accuracy.

## 3. Indoor Positioning System Based on a Digital Twin

### 3.1. System Architecture

In order to correct the NLOS error and improve the accuracy of IPS, a UWB indoor positioning system based on a digital twin is proposed in this paper as presented in Figure 1. By building a digital twin of the indoor environment and positioning system, it dynamically perceived the indoor environment and positioning target. The evolution of the model was driven by perceptual data; anchor layout optimization and error map construction were carried out to predict and eliminate the error factors in the positioning process and improve the accuracy and efficiency of the positioning system.

The system is divided into three parts: physical space, digital space, and interactive layer.

In the physical space, the spatial structure, anchor point coordinates, label coordinates, and error data in the physical space are sensed and transmitted with the digital space through wired or wireless networks.

In the digital space, the building structure entities are digitally constructed in 3D, and the error data are used to drive the error model building. The model evolution is driven by the positioning data to calculate the optimal anchor points, construct error maps, correct the positioning data, and guide the physical space decisions based on the evolution results.

The data transmission between the physical and digital layers is conducted through the interaction layer. The data sensed in the physical space are sent to the digital space through the interaction layer to drive the model evolution. The evolution results of the virtual space are fed back to the physical space through the interaction layer to guide the physical space decision making.

The specific process mainly included error model construction, optimal anchor layout, and error mitigation in several parts. Firstly, the propagation characteristics of UWB positioning system were measured in physical space, and the ranging error model and vector positioning error model of the system were constructed in digital space. The best anchor position was calculated in the digital space to guide the installation of anchors in the physical space, and then the model in the digital space was modified according to the actual installation position of anchors in the physical space. Based on the calculated positions in the digital space, measuring points were set in the physical space and data were collected. A neural network was trained using the data generated by the error model and corrected by the collected data. The data generated by the tag in the physical space will be transferred to the digital space in real time and corrected by the neural network, thus reducing the positioning error.

### 3.2. Positioning Error Model

The positioning error of the UWB system relies on the anchor layout and the tag’s distance from the anchors. The ranging error between the anchor and the tag is the main source of the positioning error. The analysis of the ranging error can be found in Appendix A.

The two-dimension UWB positioning system requires at least three anchors. Suppose the three anchors are A, B, and C, respectively, and one tag X needs to position. The coordinates of anchor A are $\left(x_{1},y_{1}\right)$, B $\left(x_{2},y_{2}\right)$, and C $\left(x_{3},y_{3}\right)$; the coordinates of the tag X are (x,y), and the distances $R_{1},R_{2},R_{3}$ denote the ranging value between the anchor point A, B, and C and the tag respectively, as follows:(1)$$\begin{array}{l} (x-x_{1}){}^{2}+(y-y_{1}){}^{2}=R_{1}{}^{2}\\ (x-x_{2}){}^{2}+(y-y_{2}){}^{2}=R_{2}{}^{2}\\ (x-x_{3}){}^{2}+(y-y_{3}){}^{2}=R_{3}{}^{2} \end{array}$$

We subtract the first equation from the last two equations in Equation (1),
(2)$$\begin{array}{l} 2x(x_{1}-x_{2})+x_{2}{}^{2}-x_{1}{}^{2}+2y(y_{1}-y_{2})+y_{2}{}^{2}-y_{1}{}^{2}=R_{2}{}^{2}-R_{1}{}^{2}\\ 2x(x_{1}-x_{3})+x_{3}{}^{2}-x_{1}{}^{2}+2y(y_{1}-y_{3})+y_{3}{}^{2}-y_{1}{}^{2}=R_{3}{}^{2}-R_{1}{}^{2} \end{array}$$

The matrix form of Equation (2) is as follows, Equation (3),
(3)$$A\left[\begin{array}{c} x\\ y \end{array}\right]=\left[\begin{array}{c} \lambda_{1}\\ \lambda_{2} \end{array}\right],$$
where
(4)$$A=\left[\begin{array}{cc} 2(x_{1}-x_{2}) & 2(y_{1}-y_{2})\\ 2(x_{1}-x_{3}) & 2(y_{1}-y_{3}) \end{array}\right],$$
(5)$$\left[\begin{array}{c} \lambda_{1}\\ \lambda_{2} \end{array}\right]=\left[\begin{array}{c} R_{2}{}^{2}-R_{1}{}^{2}+x_{1}{}^{2}-x_{2}{}^{2}+y_{1}{}^{2}-y_{2}{}^{2}\\ R_{3}{}^{2}-R_{1}{}^{2}+x_{1}{}^{2}-x_{3}{}^{2}+y_{1}{}^{2}-y_{3}{}^{2} \end{array}\right].$$

Then, the least square solution of the tag coordinates is:(6)$$\left[\begin{array}{c} x\\ y \end{array}\right]=A^{+}\left[\begin{array}{c} \lambda_{1}\\ \lambda_{2} \end{array}\right],$$
(7)$$A^{+}=A^{-1}=\frac{A^{*}}{\left|A\right|},$$
where
(8)$$A^{*}=\left[\begin{array}{cc} 2(y_{1}-y_{3}) & 2(y_{2}-y_{1})\\ 2(x_{3}-x_{1}) & 2(x_{1}-x_{2}) \end{array}\right]=2\left[\begin{array}{cc} \left(y_{1}-y_{3}\right) & \left(y_{2}-y_{1}\right)\\ \left(x_{3}-x_{1}\right) & \left(x_{1}-x_{2}\right) \end{array}\right],$$
(9)$$\left|A\right|=4[(x_{1}-x_{2})(y_{1}-y_{3})-(y_{1}-y_{2})(x_{1}-x_{3})].$$

Let
(10)$$\alpha_{1}=x_{1}{}^{2}-x_{2}{}^{2}+y_{1}{}^{2}-y_{2}{}^{2},\alpha_{2}=x_{1}{}^{2}-x_{3}{}^{2}+y_{1}{}^{2}-y_{3}{}^{2}.$$

Equation (6) can be expressed as follows:(11)$$\begin{array}{l} \left[\begin{array}{c} x\\ y \end{array}\right]=A^{+}\left[\begin{array}{c} \lambda_{1}\\ \lambda_{2} \end{array}\right]=\frac{2}{\left|A\right|}\left[\begin{array}{cc} \left(y_{1}-y_{3}\right) & \left(y_{2}-y_{1}\right)\\ \left(x_{3}-x_{1}\right) & \left(x_{1}-x_{2}\right) \end{array}\right]\left[\begin{array}{c} R_{2}{}^{2}-R_{1}{}^{2}+\alpha_{1}\\ R_{3}{}^{2}-R_{1}{}^{2}+\alpha_{2} \end{array}\right]\\ =\frac{2}{\left|A\right|}\left[\begin{array}{c} \left(y_{1}-y_{3})(R_{2}{}^{2}-R_{1}{}^{2}+\alpha_{1})+(y_{2}-y_{1})(R_{3}{}^{2}-R_{1}{}^{2}+\alpha_{2}\right)\\ \left(x_{3}-x_{1})(R_{2}{}^{2}-R_{1}{}^{2}+\alpha_{1})+(x_{1}-x_{2})(R_{3}{}^{2}-R_{1}{}^{2}+\alpha_{2}\right) \end{array}\right]\\ =\frac{2}{\left|A\right|}\left[\begin{array}{c} (y_{3}-y_{2})R_{1}{}^{2}+(y_{1}-y_{3})R_{2}{}^{2}+(y_{2}-y_{1})R_{3}{}^{2}+\beta_{1}\\ (x_{2}-x_{3})R_{1}{}^{2}+(x_{3}-x_{1})R_{2}{}^{2}+(x_{1}-x_{2})R_{3}{}^{2}+\beta_{2} \end{array}\right] \end{array}$$

In Equation (11),
(12)$$\begin{array}{l} \beta_{1}=(y_{1}-y_{3})\alpha_{1}+(y_{2}-y_{1})\alpha_{2}\\ \beta_{2}=(x_{3}-x_{1})\alpha_{1}+(x_{1}-x_{2})\alpha_{2} \end{array}$$

According to Equation (11), the positioning coordinates of the tag can be obtained. Due to the NLOS error, the ranging value from the anchor to the tag increases. The ranging error between the anchor and the tag is denoted as $\Delta_{i}$. The distribution of the ranging error is shown and analyzed in Section A.1. The ranging values of the three anchor points and the tag are $R_{1}+\Delta_{1},R_{2}+\Delta_{2},R_{3}+\Delta_{3}$, respectively. At this time, the positioning coordinates are as follows:(13)$$\left[\begin{array}{c} x^{\prime }\\ y^{\prime } \end{array}\right]=\frac{2}{\left|A\right|}\left[\begin{array}{c} (y_{3}-y_{2}){\left(R_{1}+\Delta_{1}\right)}^{2}+(y_{1}-y_{3}){\left(R_{2}+\Delta_{2}\right)}^{2}+(y_{2}-y_{1}){\left(R_{3}+\Delta_{3}\right)}^{2}+\beta_{1}\\ (x_{2}-x_{3}){\left(R_{1}+\Delta_{1}\right)}^{2}+(x_{3}-x_{1}){\left(R_{2}+\Delta_{2}\right)}^{2}+(x_{1}-x_{2}){\left(R_{3}+\Delta_{3}\right)}^{2}+\beta_{2} \end{array}\right].$$

The relationship between the positioning error and ranging error is as follows:(14)$$\begin{array}{l} \Delta_{x}=x^{\prime }-x\\ =\frac{2}{\left|A\right|}[(y_{3}-y_{2})\Delta_{1}(2R_{1}+\Delta_{1})+(y_{1}-y_{3})\Delta_{2}(2R_{2}+\Delta_{2})+(y_{2}-y_{1})\Delta_{3}(2R_{3}+\Delta_{3})]\\ \Delta_{y}=y^{\prime }-y\\ =\frac{2}{\left|A\right|}[(x_{2}-x_{3})\Delta_{1}(2R_{1}+\Delta_{1})+(x_{3}-x_{1})\Delta_{2}(2R_{2}+\Delta_{2})+(x_{1}-x_{2})\Delta_{3}(2R_{3}+\Delta_{3})] \end{array}$$

According to Equation (14), the positioning error of any point in space can be calculated from the coordinates of the anchor and the coordinates of the obstacle.

### 3.3. Anchor Layout Optimization

The anchor layout and the location of anchor points mainly influence the positioning error. When there are obstacles $O_{1},O_{2},\ldots O_{n}$ between the anchor and the tag, the transmission speed of the electromagnetic wave slows down due to passing through obstacles, which lengthens the time for the electromagnetic wave from the anchor point to reach the tag.
(15)$$t=\frac{l_{ST}}{c}+\overset{n}{\underset{i=1}{\sum }}t_{i}.$$
where $l_{st}\;$ is the actual distance between the anchor and the tag and c is the propagation speed of the electromagnetic wave in air,
(16)$$t_{i}=\frac{h_{i}\sqrt{\varepsilon_{i}}}{c}-\frac{h_{i}}{c}=\frac{h_{i}\left(\sqrt{\varepsilon_{i}}-1\right)}{c},$$
where $t_{i}$ is the time spent by the electromagnetic wave passing through the obstacle $O_{i}$ rather than propagating the same distance only in the air, $h_{i}$ and $\varepsilon_{i}$ are the thickness and dielectric constant of the obstacle $O_{i}$, respectively. This results in an increase in the measurement distance:(17)$$l_{st}{}^{\prime }=l_{st}+\overset{n}{\underset{i=1}{\sum }}t_{i}*c=l_{st}+\overset{n}{\underset{i=1}{\sum }}l_{i}.$$

Therefore, the ranging error is as follows:(18)$$\Delta_{l}=\overset{n}{\underset{i=1}{\sum }}l_{i}.$$

In the two-dimensional space, the space is divided into m∗n small square grids, and each small grid Ti is represented by the central coordinates $\left(x_{Ti},y_{Ti}\right)$; for a base station Sj with coordinates $\left(x_{Sj},y_{Sj}\right)$, it is assumed that the NLOS error in the small square grid is the same as the center point. Equation (18) is the ranging error of each base station. By substituting this into Equation (14), the positioning error of each small square grid Ti is as follows:(19)$$\Delta \left(Ti\right)=\sqrt{\Delta_{x}{}^{2}+\Delta_{y}{}^{2}}.$$
where $\lambda_{i}$ is the weight of the *i*th small square, and the sum of the spatial positioning errors is as follows:(20)$$\Delta \left(S\right)=\frac{\sum_{i=1}^{m*n}\lambda_{i}\Delta \left(Ti\right)}{m*n}.$$

Our objective function can be expressed as:(21)G=min(Δ(S)),
(22)$$s.t.\;x_{a}<x_{Sj}<x_{b},y_{a}<y_{Sj}<y_{b},z_{a}<z_{Sj}<z_{b},$$
(23)$$\Delta \left(Ti\right)<\Delta_{max},$$
(24)$$d_{s}>d_{min},d_{i}=\sqrt{{\left(x_{Si}-x_{Sj}\right)}^{2}+{\left(y_{Si}-y_{Sj}\right)}^{2}+{\left(z_{Si}-z_{Sj}\right)}^{2}}.$$

Constraint (22) represents the feasible installation area of the anchor point, Constraint (23) indicates that the positioning error of the main area in the representation space is less than the threshold $\Delta_{max}$, and Constraint (24) indicates that the distance between any two base stations is greater than the threshold $d_{min}$.

Assuming that K anchors need to be deployed, the optimal deployment location of anchors is solved according to the slime mold algorithm (SMA),
(25)$$\left(x_{S1},y_{S1}\right),\left(x_{S2},y_{S2}\right)\ldots \left(x_{SK},y_{SK}\right)=\arg\min\left(G\right).$$

The control parameter is $\left(x_{S1},y_{S1},x_{S2},y_{S2}\ldots x_{SK},y_{SK}\right)$, the number of variables is 2K, and the objective function is the fitness function. We initialize the population, randomly generate M groups of control parameters according to the constraints, and form a matrix with the size of M*2K. Then, we code each group of control parameters, and calculate the fitness of each group. We select N groups of parameters with better fitness and a smaller objective function.

After a certain number of iterations, the generated new parameters will be closer and closer to the optimal solution; hence, we select S groups of parameters with the highest fitness as the approximate optimal solution. For S groups of approximate optimal solutions, the optimal anchors deployment location are selected according to the actual installation situation and operation difficulty, and then the anchors’ location in the virtual space are corrected according to the actual installation location.

### 3.4. Positioning Error Mitigation

The positioning error model allows obtaining the positioning error distribution within an arbitrary environment, which is also known as an error map. The error map constructs the mapping from real coordinates to measurement coordinates. However, in order to eliminate errors in the localization process, it is necessary to construct an inverse mapping from measurement coordinates to true coordinates, which usually requires a large amount of data and a complex mathematical model.

An error mitigation method based on deep learning is proposed to fit the mapping of measurement coordinates to true coordinates by neural networks. The measurement coordinates containing positioning errors were converted to the predicted real coordinates, and the mitigation of positioning errors was achieved. The error map can generate a large amount of virtual data for training, and at the same time, the model can be corrected using the real measurement data, which achieves the integration of the fundamental theory and data. The above method can improve the accuracy of the positioning system and reduce the difficulty of error mitigation.

To ensure real-time computation, a lightweight network model was used. It contained three convolutional layers, one pooling layer, and four fully connected layers. A measurement coordinate is denoted as $\left[x_{i},y_{i}\right]$, and eight measurement coordinates of the same measuring point form a set of input $\left[x_{1},y_{1},x_{2},y_{2}\ldots x_{8},y_{8}\right]$ of size 1 × 16. The input data were first expanded in dimension by the fully connected layer, then resized to n × n, and passed through the convolutional and pooling layers for feature extraction. Finally, the regression calculation was performed by the fully connected layer, and the output was resized to 1 × 2. The output of the model was the real coordinates [x,y] corresponding to the measurement coordinates. The structure of the above neural network is shown in Figure 2.

According to the error model proposed in Section 3.2, the error distribution in space can be calculated, and data can be generated for the neural network to train the model. However, the actual error distribution is often more complex and variable than the predicted results, so some real positioning data are needed to correct the model. In order to obtain the real positioning data, a large number of measuring points need to be densely arranged in space. For a 5 m × 5 m space and a 10 cm interval of measuring points, 2500 measuring points are required. At the same time, in order to reduce the impact of environmental noise, each measuring point needs to collect data at least hundreds of times, which is very time-consuming.

An adaptive error map construction method was proposed in [46]. Firstly, a small number of measuring points were used to obtain the rough positioning error map, and then different measuring point densities were selected according to the size and gradient of the positioning error, so as to realize the self-adaptive error map construction process.

The error model in Section 3.2 can calculate the distribution of positioning errors, and the density of measuring points can be set directly based on the calculation results instead of using a small number of measuring points to obtain a rough error map. The space is divided into grids of different sizes to achieve different measuring point density, and each measuring point is in the central area of the grid.

According to the ranging error distribution in Section A.1, in the case of LOS, the mean and standard deviation of ranging error were small, which shows that the accuracy and stability of the data were good in the case of LOS. Therefore, using a larger grid, the density of measuring points was small; In the case of NLOS, the error and standard deviation of ranging data increase significantly, which will lead to the significant increase and instability in the positioning error, which indicates that the change in positioning error is more intense. Therefore, a smaller grid was used, and the density of measuring points was larger.

For the area not covered by obstacles, the positioning error was usually within 10 cm, and the error was relatively small. This is considered a high precision area with a grid size of 10 cm × 10 cm. For the area covered by obstacles, the localization error was relatively large. This is considered a low accuracy area with a grid size of 5 cm × 5 cm.

## 4. Experimental Results

The experimental environment was a rectangular area with a length of 1830 cm and a width of 840 cm as shown in Figure 3. There were eight solid concrete columns with a length and width of 70 cm as obstacles, which are labeled 1 to 8 in the figure. A UWB positioning system based on dwm1000 was adopted, and its ranging and positioning resolution was 1 cm.

The vector positioning error in space is shown in Figure 4; the blue triangles represent the anchors, and the red boxes represent the obstacles. The resolution of the prediction error was 10 cm, and the positioning error of the center position of the small square grid with a side length of 10 cm represented the error distribution in the grid. The coordinates of the three base stations were (380, 0), (940, 720), (1440, 10), respectively. It can be seen from the figure that, in the area not covered by obstacles, the positioning error was small, usually less than 10 cm; there is a positioning error of 10–50 cm in the slightly obscured area, which means that the obstruction was not very serious; in some areas with serious occlusion, the positioning error was more than 50 cm, which is usually caused by thicker obstacles.

In the experimental part, the accuracy of the vector positioning error model was first checked to verify whether it could approximately describe the positioning error distribution in space. Next, several sets of optimal anchor locations were solved with the optimal anchor layout algorithm and compared with several sets of anchor locations set empirically. The sum of errors in the space was compared to check the effectiveness of the algorithm. Then, the number of measuring points with adaptive density in different experimental scenarios was counted and compared with the number of measuring points with fixed density. Finally, a neural network model was trained using the virtual data and measurement data, and we tested whether the model could effectively eliminate the localization error.

### 4.1. Validation of Positioning Error Model

In order to verify the accuracy of the vector positioning error model, we set measuring points in different areas, moved the tag to each measuring point, collected 500 groups of positioning data at each measuring point, and compared with the prediction error to verify the accuracy of the error model.

Figure 5a–c show the data of three sets of measurement points with coordinates (100,450), (100,150), (900,450), respectively, which were selected from different areas in the experimental scenario. The red “+” indicates the 500 sets of measurement coordinates, the blue dot indicates the mean value of these 500 sets of coordinates, the black arrow is the vector positioning error around the measurement point calculated according to the error model, and the yellow dot indicates the predicted measurement coordinates based on the surrounding calculation error. The distance between the mean value of the measured coordinates and the true and predicted coordinates decreased from 131 to 43 cm, as shown in Figure 5a, from 223 to 111 cm, as shown in Figure 5b, and as shown in Figure 5c, the values were essentially the same because of the small positioning error. It can be seen from the figure that the predicted coordinates were closer to the measured coordinates in different regions, which indicates that the vector positioning error model proposed in Section 3.2 could effectively predict the positioning error distribution in space.

### 4.2. Validation of Anchor Layout Optimization

In order to verify whether the study on the anchor layout optimization could effectively reduce the positioning error in space, the optimal anchor layout solved using the algorithm was compared with the empirical anchor layout. The slime mold algorithm (SMA) was used to obtain the optimal anchor layout. The number of variables was six, the number of population was set to 50, and the number of iterations was set to 80 in this SMA. Figure 6 shows the distribution of positioning errors under one possible optimal anchor layout.

Figure 7a shows the comparison of the positioning errors of the anchor position obtained by SMA and the anchor position placed according to experience. The blue part in the histogram is the sum of the spatial positioning errors of the four groups of anchor positions placed according to experience, and the orange part is the anchor position obtained by SMA. Through comparison, it can be seen that after SMA optimization, the total spatial positioning error was significantly reduced from 250,000 cm to 200,000 cm, with a decrease of at least 25%, which proves the effectiveness of SMA. Figure 7b is a comparison of the sum of the positioning errors in different regions in the space. The solid line with triangles in the figure represents the layout using SMA, and the dotted line with “x” represents the layout according to experience. It can be seen from the figure that the anchor points placed according to experience in a small part of the region had better error performance. However, on the whole, the use of SMA effectively reduced the positioning error and achieved a better anchor layout. At the same time, SMA greatly reduced the calculation time, so as to complete the layout of anchor points in a short time.

### 4.3. Positioning Error Mitigation

During the construction of the adaptive error map, in order to select the appropriate grid size, two groups of measuring points were selected in an LOS scene and an NLOS scene, respectively, with a total of 25 measuring points in five rows and five columns in each group. Each measuring point collected 500 groups of positioning data, and the starting coordinates of the two groups of measuring points were (920,300) and (550,500), respectively. The positioning results of the two groups of measuring points are shown in Figure 8. Each measuring point in the figure is represented by dots of different colors, the 500 positioning data of the same measuring point are represented by “+” of the same color, and the average value of the measured data is represented by triangles of the same color. As can be seen from Figure 8a, the positioning accuracy in the LOS scene was relatively high, the error was usually within 10 cm, and the multiple positioning coordinates of each measuring point were relatively more concentrated with smaller standard deviation. The positioning errors between two adjacent measuring points were similar, so the error distribution could be described more accurately by using the measuring point interval of 10 cm. As shown in Figure 8b, the positioning error and standard deviation in the NLOS scene were larger. The positioning error was usually more than 1 m. The positioning results of the same measuring point were scattered, and the positioning results of two adjacent measuring points were also quite different. Therefore, a smaller interval of measuring points should be used in the NLOS scenario. The positioning accuracy of the UWB system itself is on the centimeter level. Using a smaller measuring point interval will not only increase the number of measuring points but can also improve the accuracy of the error map. Finally, 5 cm was selected as the measuring point interval in NLOS scene.

At the same time, due to the adaptive grid size, the number of measuring points can be greatly reduced while ensuring the accuracy of the error map. All used an interval of 10 cm, and the number of measuring points was 14,400. However, using the adaptive grid size, only 6300 measuring points were required, and the number of measuring points was reduced by 56%, which greatly reduced the difficulty of constructing the error map. The adaptive measuring point density schematic for the same experimental scenario is shown in Figure 9.

To mitigate the positioning error, the neural network model proposed in Section 3.4 needs to be trained. The model was pretrained by generated virtual data based on the error model, and then the model was corrected using the real measurement data. The virtual data were generated at 10 cm intervals; the experimental scene contained 185 × 85 points, each point generated 20 × 8 groups of data, and each group contained the real coordinates [x,y] and the predicted measurement coordinates $\left[x_{i},y_{i}\right]$, as shown in Figure 10. The eight sets of measurement coordinates at the same point were used as input, and the corresponding real coordinates were used as output to train the neural network. Based on the model trained on the virtual data, the model was modified by retraining using the two sets of real measurement data mentioned above. There were 25 measuring points in the LOS and NLOS scenarios, respectively, and each measuring point contained 500 sets of data, 496 (62 × 8) sets of data were selected to form the data set. Of these, 80% were used as the training set and 20% as the test set.

Figure 11a,b show the error mitigation effects for the LOS and NLOS test sets, respectively. Different colors are used to distinguish the measuring points with different coordinates, the dots represent the true coordinates of the measuring points, the triangles represent the measured coordinates, and the stars represent the predicted coordinates output after feeding the measured coordinates into the model. Most of the predicted coordinates in the figure were closer to the true coordinates than the measured coordinates, which shows that our model can effectively mitigate the positioning errors. Figure 11c,d respectively show 100 points in the test set under the LOS and NLOS scenarios. The line with “x” represents the original error of the measured coordinates, the line with “. “ represents the error corrected by the neural network model, and the line with triangles represents the error corrected by the cubic polynomial surface fit. The average localization error was reduced from 3.64 cm to 2.49 cm in the LOS and from 72.92 cm to 6.69 cm in the NLOS. As a comparison, the error mitigation using the conventional cubic polynomial surface fitting approach only reduced to 42.79 cm in NLOS but increased to 5.47 cm in LOS instead. This is due to the fact that the error in the LOS was so small that the polynomial failed to fit the functional relationship better. The above experimental results indicate that after the error mitigation by the neural network, the accuracy of the UWB positioning system was improved in both the LOS and NLOS cases.

## 5. Conclusions

In this work, the ranging data of a single anchor single tag UWB system in an LOS scenario and an NLOS scenario were collected, and the ranging error distribution of a UWB system was established by the probability density function of Gaussian distribution. The vector positioning error model of the UWB positioning system was obtained by mathematical derivation. In addition, the SMA was introduced to optimize the anchor point position during the deployment of the UWB system. A deep learning-based error mitigation method was used to improve the positioning accuracy of the UWB system in the LOS and NLOS scenarios. The experimental results showed that the proposed vector positioning error model accurately described the positioning error distribution in space, and the anchor location optimization algorithm reduced the total spatial positioning error by about 25–40%, which greatly reduced the calculation time compared with the exhaustive method. Using the method of adaptive measuring point density to construct the error map reduced the number of measuring points by about 56%. The neural network-based localization error mitigation method reduced the average localization error from 3.64 cm to 2.49 cm in the LOS scenarios and from 72.92 cm to 6.69 cm in the NLOS scenarios. In future research, we will further study the propagation of UWB signals in different situations, the layout method of multiple anchors, and the correction of positioning coordinates in the motion state.

## Figures and Tables

**Figure 1 sensors-22-05936-f001:**
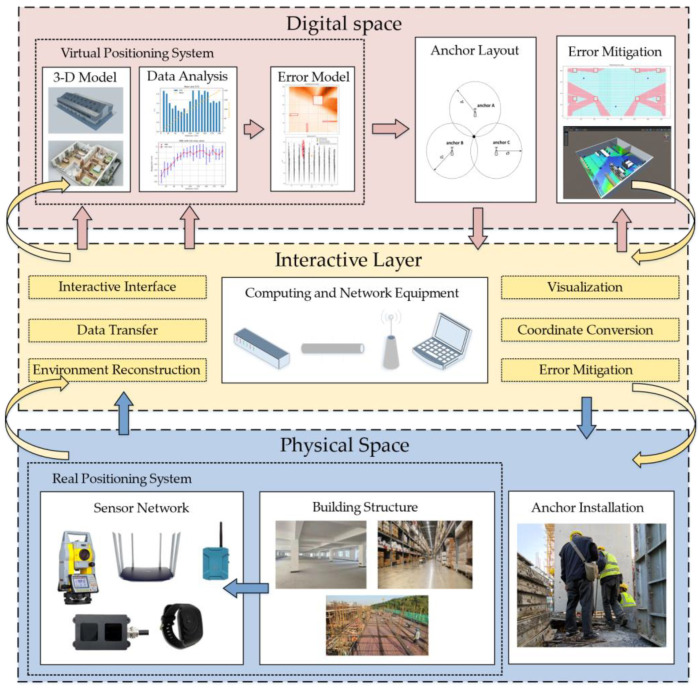
Architecture of the system.

**Figure 2 sensors-22-05936-f002:**
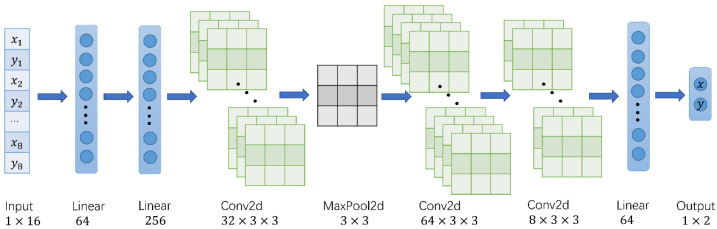
Neural network structure diagram.

**Figure 3 sensors-22-05936-f003:**
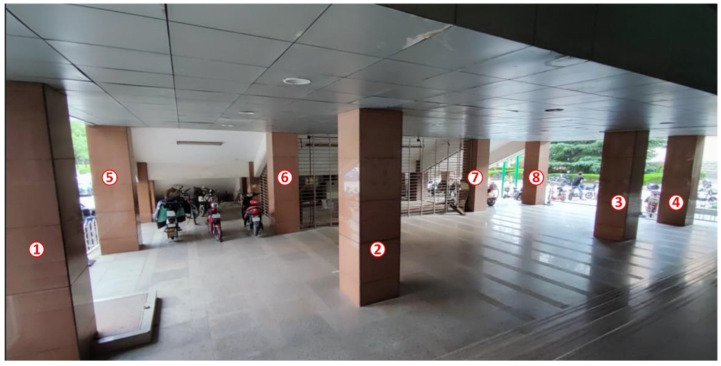
Photograph of the real experimental environment.

**Figure 4 sensors-22-05936-f004:**
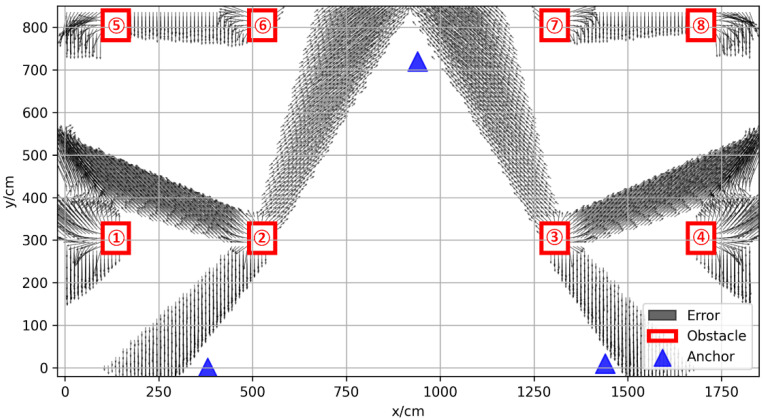
Vector positioning error map of experimental environment.

**Figure 5 sensors-22-05936-f005:**
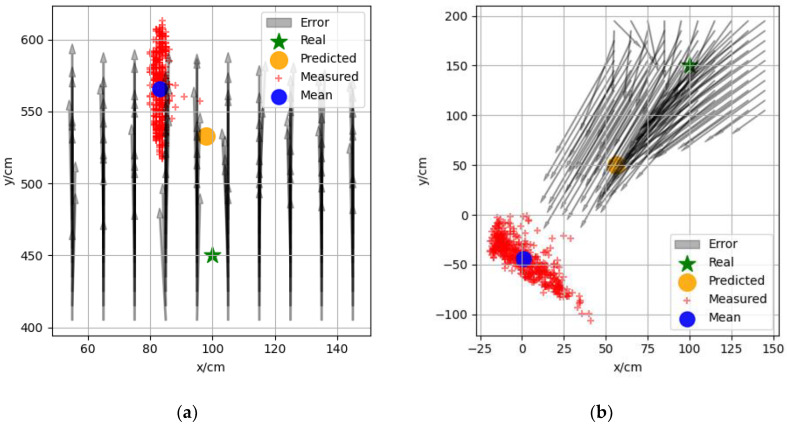
Real coordinates, measurement coordinates, and prediction coordinates of measuring points. Real coordinates of the measuring points: (**a**) (100,450); (**b**) (100,150); (**c**) (900,450).

**Figure 6 sensors-22-05936-f006:**
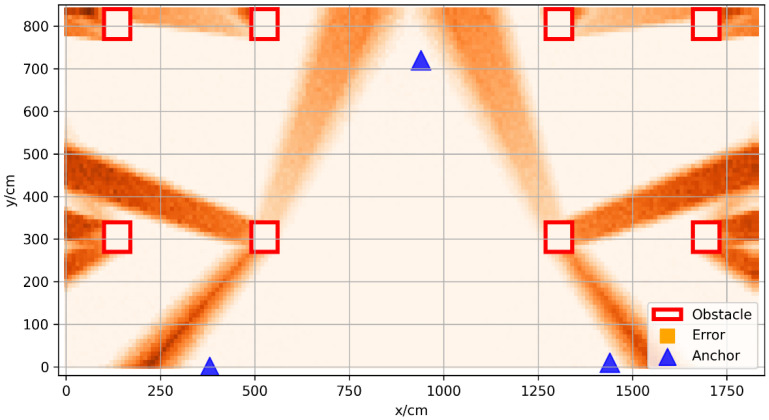
The approximately optimal anchor layout.

**Figure 7 sensors-22-05936-f007:**
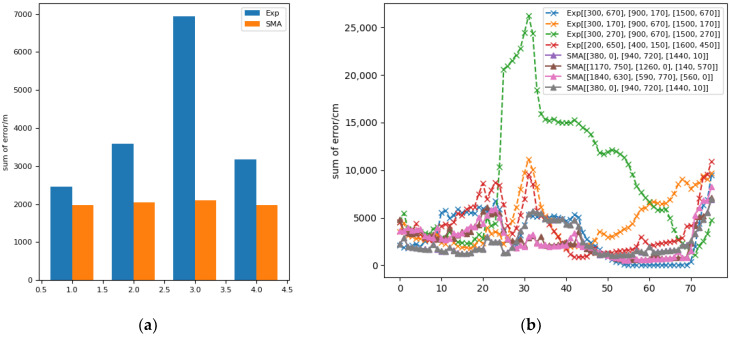
(**a**) Comparison of the spatial positioning error between the anchor position optimized by SMA and the anchor position placed by experience (Exp). (**b**) Comparison of positioning errors in different regions.

**Figure 8 sensors-22-05936-f008:**
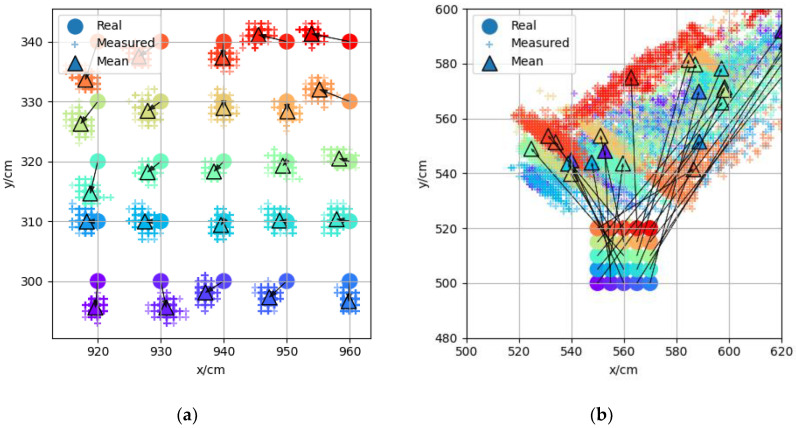
(**a**) Measuring point group (920,300) in the LOS scene. (**b**) Measuring point group (550,500) in the NLOS scene.

**Figure 9 sensors-22-05936-f009:**
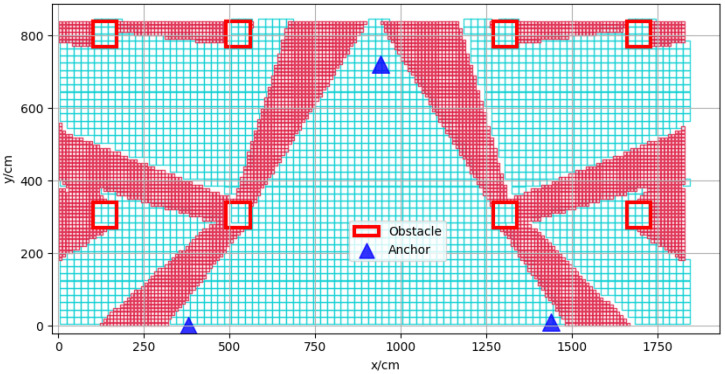
Adaptive measuring point density diagram.

**Figure 10 sensors-22-05936-f010:**
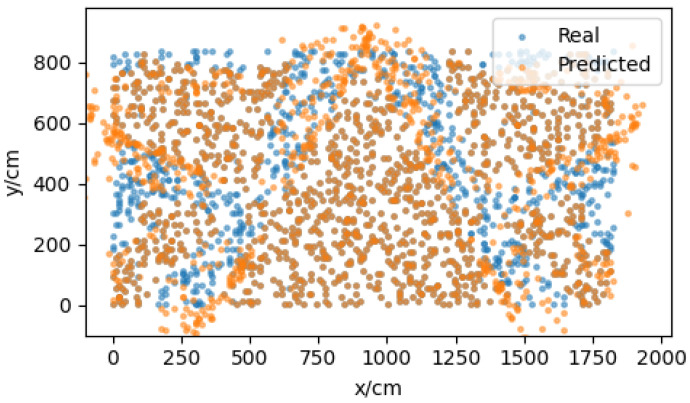
Part of the virtual data generated by the vector positioning error model.

**Figure 11 sensors-22-05936-f011:**
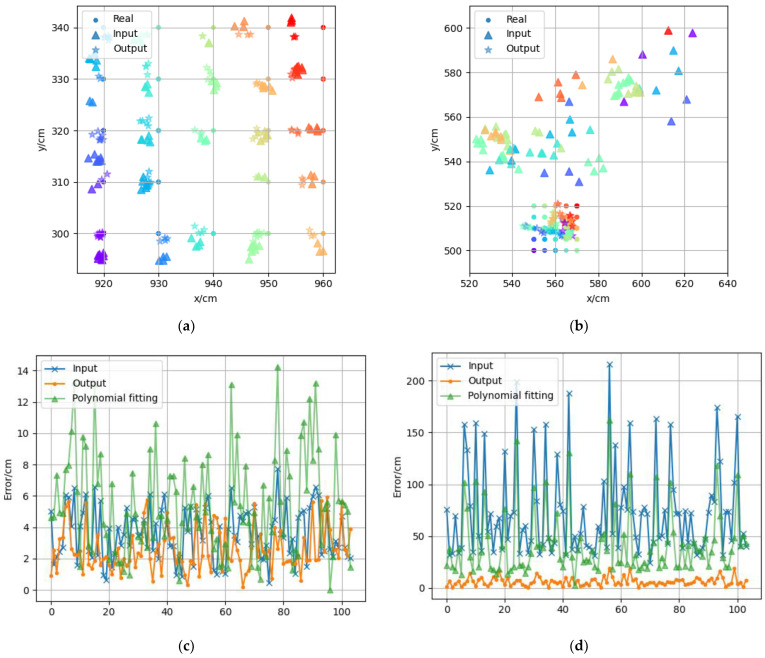
Results of Positioning Error Mitigation in LOS (**a**) and NLOS (**b**). Comparison of Error Mitigation Results for Neural Networks and Polynomial Fitting in LOS (**c**) and NLOS (**d**).

## Data Availability

Not applicable.

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
