# Peer review of "Indoor Positioning System with UWB Based on a Digital Twin"

_sensors, 2022, doi:10.3390/s22165936_

Round 1
Reviewer 1 Report
Overall, this is an interesting study and the work is well presented. It will contribute to the field if some parts could be improved. Some comments:
1. In line 55, "Positioning error elimination method" is mentioned, but the description in other parts is "Positioning error mitigation method", I suggest to change them to be consistent.
2. In sections 3.2 and 3.3, the authors conducted a study on "Positioning error model" and "Optimal anchor layout", respectively. I suggest that the authors describe the purpose and significance of the study in more detail.
3. The description of Figure 3 is "vector positioning error map", but the error is not shown in the figure, so the authors should replace the description.
4. The analysis of the experimental results in section 4.1 is not sufficient and lacks data support and the graphs of experimental results are missing units of coordinates. I suggest that the authors provide a more detailed analysis of the experimental results.
5. Part of Figure 7(b) is obscured by the legend. I suggest that the authors redraw this figure to show the full experimental data.
6. The purpose of the experiment should be presented in each section of the experimental results in Section 4.
7. There are some grammatical errors, please check carefully.
Author Response
- In line 55, "Positioning error elimination method" is mentioned, but the description in other parts is "Positioning error mitigation method", I suggest to change them to be consistent.
Thank you very much for your suggestion. "Positioning error elimination method" is replaced with "Positioning error mitigation method" in the manuscript.
- In sections 3.2 and 3.3, the authors conducted a study on "Positioning error model" and "Optimal anchor layout", respectively. I suggest that the authors describe the purpose and significance of the study in more detail.
Thank you very much for your suggestion. We focus on how the ranging error causes the positioning error in Section 3.2 in order to calculate the positioning error based on the ranging error. In Section 3.3, we focus on the effect of anchor layout on the localization error in space and summarize the optimization objective as an objective function. We have added the purpose and significance of the study at the beginning of each section.
- The description of Figure 3 is "vector positioning error map", but the error is not shown in the figure, so the authors should replace the description.
Thank you very much for your suggestion. Figure 3 is a photograph of the real experimental environment and Figure 4 is a simulation of the vector positioning error, we have changed the description of Figure 3 to "Photograph of the real experimental environment."
- The analysis of the experimental results in section 4.1 is not sufficient and lacks data support and the graphs of experimental results are missing units of coordinates. I suggest that the authors provide a more detailed analysis of the experimental results.
Thank you very much for your suggestion. We have added further analysis of the experimental results in Section 4.1 and provided comparisons on the data to further validate the accuracy of the positioning error model.
- Part of Figure 7(b) is obscured by the legend. I suggest that the authors redraw this figure to show the full experimental data.
Thank you very much for your suggestion. We have modified Figure 7(b) by adjusting the position of the legend to avoid obscuring the content.
- The purpose of the experiment should be presented in each section of the experimental results in Section 4.
Thank you very much for your suggestion. The content in Section IV focuses on the experimental validation of the model and algorithm proposed in Section III, and the purpose of the experimental part we have described in line 325 of Section IV. To make the content clearer, we describe it in more detail in Section 4.2.
- There are some grammatical errors, please check carefully.
Thank you for your suggestion, we have carefully checked the paper for grammatical errors and fixed them.
Reviewer 2 Report
The authors present an interesting paper about indoor position system based on UWB digital twin. However they should improve some issues, namely:
1.- a rview of written english is advisable, as the document has some typos;
2.- the authos refer to measurement error typically from a qualitative point of view. They do not give a concrete explanation as to its origin and what clearly contributes to it. Also, they do not comment on the behavior of the error in relation to the distance to the anchors used. They refer to "slightly obscured areas" without specifying very well what they mean. Does it mean that at least one of the anchors is not in LOS? Furthermore, it is not clear what the contribution of the measures of each of the anchors is it the same for all? do you use different weights for each one? the distance to each of them of them how does it interfere in the model? For instance, in all the cases studied where one of the anchors is NLOS, is the error magnitude of the same dimension in the different cases?
Author Response
- a review of written english is advisable, as the document has some typos;
Thank you very much for your suggestion. The written English has been checked again and corrected in the paper.
- the authors refer to measurement error typically from a qualitative point of view.
2.1 They do not give a concrete explanation as to its origin and what clearly contributes to it.
Thank you very much for your suggestion. The errors in this paper are mainly divided into positioning errors and ranging errors. The positioning coordinates are obtained from the ranging values, so the positioning error is mainly caused by the ranging error. The detail is in Section 3.2 and added more explanations. The "measurement error" mentioned in the appendix indicates the ranging error and is replaced by "ranging error". The ranging error comes from the increasing of the electromagnetic wave flight time due to obstruction, which is explained in detail in Section 3.3 (highlight).
2.2 Also, they do not comment on the behavior of the error in relation to the distance to the anchors used.
Thank you very much for your suggestion. In Section 3.2, we studied the influence of ranging error on positioning error. The distance from the tag to the anchor is denoted as "", Where "" represents this ranging error, and Equation (14) shows the relationship between ranging error and positioning error. The analysis of the ranging error is carried out in the appendix, where we focus on the error between the true and measured distances of anchor and tag at different distances.
2.3 They refer to "slightly obscured areas" without specifying very well what they mean. Does it mean that at least one of the anchors is not in LOS?
Thank you very much for your suggestion. We mentioned "slowly observed area" in line 321 of section 4, which means that the positioning error is between 10-50 cm. This is not related to the number of anchors in NLOS, because the thickness of the obstacle is also an important factor in the localization error, and the degree of obscured area cannot be judged only by whether the anchor is in LOS or not. We have described the " slowly observed area " more precisely in the paper.
2.4 Furthermore, it is not clear what the contribution of the measures of each of the anchors is it the same for all? do you use different weights for each one?
Thank you very much for your suggestion. The effect of ranging error from different anchors on the positioning error is studied in Section 3.2. From Equation (14), we can see that the effect of ranging error from different anchors is mainly reflected in the direction of the positioning error, which is the reason why we use vector to represent the positioning error. Different anchors have the same weight in the current study, which is due to the relatively small number of anchors, and we may consider using different weights for different anchors in future studies.
2.5 The distance to each of them of them how does it interfere in the model? For instance, in all the cases studied where one of the anchors is NLOS, is the error magnitude of the same dimension in the different cases?
Thank you very much for your suggestion. The effect of the range value of each anchor on the localization error can be seen from Equation (14) in Section 3.2, where the range errors of different anchors are expressed as "" and ultimately cause the localization error "". However, the positioning error is not only related to the ranging error of the anchor, but also related to the coordinates of the anchor itself and the coordinates of the tag, so the magnitude of the error is different in different cases.

Author Response
- The labeling of x and y axis is needed in all figures.
Thank you very much for your suggestion. all the figures have been corrected according to your suggestion.
- Authors claimed that “A deep learning-based error mitigation methods can effectively reduce the positioning error of UWB system and improve positioning accuracy”. But they did not include any comparison with the existing solutions.
Thank you very much for your suggestion. We add a comparison with the error mitigation method based on surface fitting in Section 4.3. for verifying the effectiveness of the deep learning-based localization error mitigation method proposed in this paper. The results show that our method has a better error mitigation effect.
3. Which physical entities they considered as input features for training the neural network? What is the ranging technique for localization?
Thank you very much for your suggestion. In Section 3.4 of this paper, an error elimination method based on deep learning is proposed, using a lightweight neural network. The purpose of this neural network is to correct the measured coordinates, so eight sets of measured coordinates and one set of real coordinates of the same point are used as input features for training.
The ranging technique used in this paper is DS-TWR, which calculates the distance between anchors and tags by using two round-trip times between them, which is a well-established ranging algorithm, and we have added a description of the ranging technique in the Appendix.
4. There are 25 measuring points in LOS and NLOS scenarios respectively, and each measuring point contains 500 sets of data, and 496(62*8) sets of data are selected to form the data set. What are the reasons to selecting 496 sets of data among 500?
Thank you very much for your suggestion. The neural network used in this paper uses eight sets of measurements as input, so the number of input data must be a multiple of eight. We collected 500 sets of data at each point, and to make full use of the collected data, the maximum multiple of 8 below 500, 496, was chosen as the number of input data.
5. The virtual data are generated at 10 cm intervals, and the experimental scene contains 185*85 points, each point generates 20*8 groups of data, each group contains the real coordinates [x, y] and the predicted measurement coordinates [xi, yi], as shown in Figure 10. Author should use the proper formatting for subscript, i.e. .
Thank you very much for your suggestion. We have revised this content and made changes to other similar issues in this paper.
Round 2
Reviewer 3 Report
current version can be published